# Investigating the Systems-Level Effect of *Pueraria lobata* for Menopause-Related Metabolic Diseases Using an Ovariectomized Rat Model and Network Pharmacological Analysis

**DOI:** 10.3390/biom9110747

**Published:** 2019-11-18

**Authors:** Ji Hong Oh, Seon-Eun Baek, Won-Yung Lee, Ji Yun Baek, Tuy An Trinh, Do Hwi Park, Hye Lim Lee, Ki Sung Kang, Chang-Eop Kim, Jeong-Eun Yoo

**Affiliations:** 1Department of Physiology, College of Korean Medicine, Gachon University, Seongnam 13120, Korea; jihong421@hanmail.net (J.H.O.); wonyung21@naver.com (W.-Y.L.); 2Department of Obstetrics and Gynecology, College of Korean Medicine, Daejeon University, Daejeon 35235, Korea; dreamingmong@naver.com; 3Department of Preventive Medicine, College of Korean Medicine, Gachon University, Seongnam 13120, Korea; wldbsttn@naver.com (J.Y.B.); tuyantrinh@gmail.com (T.A.T.); parkdo@gachon.ac.kr (D.H.P.); kkang@gachon.ac.kr (K.S.K.); 4Department of Food Science, Gyeongnam National University of Science and Technology, Jinju 52725, Korea; 5Department of Pediatrics, College of Korean Medicine, Daejeon University, Daejeon 35235, Korea; hanilim@dju.ac.kr

**Keywords:** menopause-related metabolic diseases, menopause, *Pueraria lobata*, network pharmacology, lipid metabolism, dyslipidemia

## Abstract

This study was conducted to evaluate the biological activities of *Pueraria lobata* (PL) on menopause-related metabolic diseases and to explore the underlying mechanism of PL by network pharmacological analyses. We used ovariectomized (OVX) rats as a postmenopausal model and administered PL at different doses (50, 100, and 200 mg/kg). In OVX rats, decreased uterine weights and PPAR-γ (peroxisome proliferator-activated receptor-gamma) mRNA expression in the thigh muscle were significantly recovered after PL administration. PL also significantly alleviated OVX-induced increases in total cholesterol, triglyceride, alanine aminotransferase (ALT/GPT), and aspartate aminotransferase (AST/GOT) levels. To identify the systems-level mechanism of PL, we performed network pharmacological analyses by predicting the targets of the potential bioactive compounds and their associated pathways. We identified 61 targets from four potential active compounds of PL: formononetin, beta-sitosterol, 3’-methoxydaidzein, and daidzein-4,7-diglucoside. Pathway enrichment analysis revealed that among female sex hormone-related pathways, the estrogen signaling pathways, progesterone-mediated oocyte maturation, oxytocin signaling pathways, and prolactin signaling pathways were associated with multiple targets of PL. In conclusion, we found that PL improved various indicators associated with lipid metabolism in the postmenopausal animal model, and we also identified that its therapeutic effects are exerted via multiple female sex hormone-related pathways.

## 1. Introduction

Menopause is a period when women’s ovarian function gradually declines and female hormones fluctuate and decrease, resulting in irregular menstruation and menopause-related symptoms. Lack of estrogen induces imbalance of energy homeostasis, by assisting energy intake and lipogenesis, depressing energy expenditure, and aggravating insulin secretion and sensitivity [1]. These metabolic changes in menopause mediate the development of chronic diseases such as cardiovascular diseases (CVD), obesity, hyperlipidemia, and fatty liver [2]. The prevalence of the metabolic syndrome in postmenopausal women is higher than in premenopausal women, which is estimated at 31–55% according to the various studies [3,4,5,6]. In postmenopausal life, prevention and management of these chronic diseases is critical for maintaining good health.

Hormone replacement therapy (HRT) is used as the first choice for treating different menopausal disabling symptoms. While HRT is effective in alleviating most postmenopausal symptoms [7], many women refuse HRT concerning the latent adverse effects and prefer the use of nonhormonal therapies to relieve symptoms after menopause [7,8]. As one of the nonhormonal treatments, natural products are considered as an alternative to menopausal symptoms and metabolic diseases caused by menopause. In particular, herbs have been widely used to relieve symptoms after menopause in East Asia, especially in Korea, China, and Japan [9,10]. Herb-based therapies were found to be associated with more reductions in the symptoms in menopausal women than in the placebo group in meta-analysis [11]. In a randomized controlled trial, whole soy had a beneficial effect on cardiovascular biomarkers in equol-producing postmenopausal women when compared with the placebo group [12]. Several experimental studies reported that traditional herbal prescriptions reduced the serum lipid levels and hepatic lipid accumulation in an ovariectomized rat model [13,14].

*Pueraria lobata* (PL), the essential herb in traditional Korean medicine, is used for the treatment of various diseases such as diabetes, liver diseases, CVD, and postmenopausal osteoporosis [15,16,17,18]. PL extracts have also been reported to prevent obesity and improve glucose metabolism [19]. *Pueraria lobata* is expected to function as an alternative to estrogen because it contains isoflavones—for instance, puerarin, daidzein, daidzin, genistin, and genistein—which are representative phytoestrogens that show structural or functional similarity to estrogen [20,21].

Network pharmacology, the novel approach based on systems biology, enables the elucidation of the potential mechanism of multiple compounds and the identification of pathways associated with the targets of the compounds at the systems level [22]. This approach can also be applied to the study of diseases affecting various systems of the body (e.g., menopause-related metabolic diseases). Since PL is composed of multiple compounds, including phytoestrogens, this would be a proper method to investigate the mechanism of PL in menopause-related metabolic diseases.

Here, we aimed to investigate the potential benefits of PL on menopause-related metabolic diseases at the systems level. To evaluate the biological activity of PL, we assessed changes in uterine weight, serum lipids, alanine aminotransferase (ALT/GPT), aspartate aminotransferase (AST/GOT), liver weight, and peroxisome proliferator-activated receptor gamma (PPAR-γ) messenger RNA (mRNA) expression in an ovariectomized (OVX) rat model. To explore the underlying mechanism of PL, we investigated target genes and pathways of PL related to menopause-related metabolic diseases by applying network pharmacology analysis (Figure 1).

## 2. Materials and Methods

### 2.1. Preparation of the Herbal Formulation

The PL (harvested from Gyeongnam, Geochang, Republic of Korea) was purchased from Okchundang Herbs (Seoul, Korea). *Pueraria lobata* (500 g) was extracted with 80% MeOH (1.5 L) three times at 25 °C, and 103.3 g of PL extract was obtained after drying under reduced pressure. The powdery dry extract of clomidin tablets (purchased from Seoul Pharmaceutical Co., Seoul, Republic of Korea), after the film coating composed of the dried extract of red clover (RC) was removed, was used as a positive control.

### 2.2. Experimental Animals and Experimental Design

We followed the Guidelines for Animal Experimentation approved by Gachon University in this study (GIACUC 2017-034). The experiment was conducted on 46 female Sprague-Dawley rats, five weeks old and weighing 130–150 g (Doo Yeol Biotech, Seoul, Korea). Housing conditions were maintained at 24 ± 2 °C; relative humidity was between 50% and 55%. Food and water were available ad libitum. After one week of acclimatization, all rats were anesthetized with an intraperitoneal injection of Zoletil^®^ (Virbac, Carros, France) and Rompun^®^ (TS, Bayer, Leverkusen, Germany) mixture (3:1, 0.2 mg/kg body weight). Sham operation was performed for six rats (sham), and bilateral ovariectomies were performed for the remaining 40 rats. After surgery, rats were fed a solid diet for one week of recovery. Ovariectomized rats were divided into five groups (*n* = 8): (1) OVX; (2) OVX + RC; (3) OVX + PL50; (4) OVX + PL100; and (5) OVX + PL200. The animals of the OVX + RC group were orally administered 3.3 mg/kg red clover extract. The animals in the OVX + PL50, OVX + PL100, and OVX + PL200 groups were orally treated with PL in aqueous solution at 50, 100, and 200 mg/kg bw, respectively. The animals in the sham and OVX groups were administered the same volume of distilled water for eight weeks. To investigate the effect of PL on menopause-related metabolic diseases, we collected blood, uterine, liver, and thigh muscle samples from each animal model.

### 2.3. Blood and Organ Dissection

At the end of the experimental period, the blood samples were collected via cardiac puncture. The uterus and liver were excised from the experimental animals and washed with saline, and the excess moisture was removed and weighed. The thigh muscles were dissected and stored in the freezer for the measurement of PPAR-γ mRNA expression.

### 2.4. Serum Analyses

Blood samples were collected in test tubes containing 0.18 M ethylenediaminetetraacetic acid (EDTA) and centrifuged at 3,500 revolution per minute (rpm) for 10 min at 4 °C. The total cholesterol (range: 3.86–800 mg/dL), high-density lipoprotein (HDL) cholesterol (range: 3–120 mg/dL), triglyceride (detection range: 8.85–885 mg/dL), AST (detection range: 5–700 U/L), and ALT (detection range: 5–700 U/L) were analyzed using the appropriate enzymatic colorimetric kits in accordance with the manufacturer’s instructions (Roche, Mannheim, Germany). The serum levels of total cholesterol (Catalog No. 05168538190, Roche), HDL (Catalog No. 05168805190, Roche), triglycerides (Catalog No. 05171407 190, Roche), AST (Catalog No. 05850819190, Roche), and ALT (Catalog No. 05850797190, Roche) were measured using a kit (Roche).

### 2.5. Measurement of PPAR-γ mRNA Expression

#### 2.5.1. Tissues and Total RNA Extraction

The thigh muscles were separated, placed in a nitrogen tank for rapid freezing, and then stored at −80 °C until use. The homogenized muscle tissue was mixed with 200 μL of chloroform (Sigma Co, St. Louis, MO, USA) per 1 mL of TRIzol solution (Favorgen Biotech Corp., Kaohsiung, Taiwan) for RNA extraction and centrifuged (13,000 rpm, 10 min, 4 °C) to collect RNA from the supernatant.

#### 2.5.2. cDNA Preparation

The RNA, DEPC (diethyl pyrocarbonate), and reverse transcriptase and Oligo (dT) 15 were mixed in tubes for polymerase chain reaction (PCR). Thereafter, the reaction was carried out at 75 °C for 5 min using a PCR machine and a cDNA synthesis kit (Intron Biotechnology, Seoul, Korea). Then, the reaction mixture was reacted at 75 °C for 5 min, and 50 μL of DEPC water was added to the obtained cDNA to be used for real time polymerase chain reaction (RT-PCR).

#### 2.5.3. Real-Time Polymerase Chain Reaction

The synthesized cDNA (1 μL) and the PPAR-γ primer (2 μL) were put into a master mix solution (Intron biotechnology), and 17 μL of DEPC water was added, followed by amplification by PCR. The PCR products were then electrophoresed on 2% agarose gels, stained with ethidium bromide, and checked with an ultraviolet (UV) lamp (NEOScience, Seoul, Korea). After PCR, the amount of PPAR-γ was calculated using an image station (Kodak, Rochester, USA) apparatus (Table 1).

### 2.6. Network Pharmacological Analysis

The PL compounds were retrieved from the Traditional Chinese Medicine Systems Pharmacology (TCMSP, http://tcmspw.com/tcmsp.php) database [23]. Potential bioactive compounds were selected from compounds with an oral bioavailability (OB) ≥30 and drug-likeness (DL) index ≥0.18, which were the default values suggested in the TCMSP [24]. We predicted potential bioactive compound targets from the TCMSP and further analyzed related signaling pathways of predicted targets using the Kyoto Encyclopedia Genes and Genomes (KEGG) database [25].

To understand the action of PL’s bioactive compounds on menopausal symptoms, we retrieved menopause-related genes from the Entrez Gene database (ncbi.nlm.nih.gov/gene) [26]. Then, we identified common targets between predicted targets from the TCMSP and retrieved targets from the Entrez Gene database. After integrating information about potential bioactive compounds of PL, their targets, and related pathways, we constructed a network graph using Cytoscape 3.5.1 (http://cytoscape.org) [27].

### 2.7. Statistical Processing

Data were analyzed using the two-tailed Mann—Whitney U test and the hypergeometric test. Statistical significance was set at the level of *p* ≤ 0.05. Bonferroni correction was used for multiple comparisons. All statistical analyses were processed using the Scipy (http://scipy.org) and Numpy (http://Numpy.org) module in Python 3.6 (http://www.python.org).

## 3. Results

### 3.1. Animal Experiments

We used OVX rats as a postmenopausal model and administered PL at different doses (50, 100, and 200 mg/kg).

#### 3.1.1. Uterine Weight

The uterine weight in the OVX group showed a dramatic decrease compared to the sham group (*p* = 0.002, Figure 2). It was observed that the PL treatment dose-dependently improved uterine weight in the OVX rats (*p* = 0.008, 0.0009, and 0.0009 per dose, respectively), which is consistent with RC administration as a positive (*p* = 0.0009).

#### 3.1.2. Serum Lipids

We examined the levels of total cholesterol, triglyceride, and HDL (Figure 3). Prominent increases in serum lipids were noted in the OVX rats when compared with sham rats (*p* = 0.002 for total cholesterol; *p* = 0.008 for triglyceride; *p* = 0.008 for HDL). Administration of 100 mg/kg PL resulted in a statistically significant reduction in total cholesterol (*p* = 0.007, Figure 3a) and triglyceride (*p* = 0.001, Figure 3b), which were consistent with the positive control (*p* = 0.0007 for total cholesterol; *p* = 0.0009 for triglyceride). However, the levels of HDL showed no significant changes in all groups treated with PL (Figure 3c).

#### 3.1.3. Glutamate Oxaloacetate Transaminase (GOT), Glutamate Pyruvate Transaminase (GPT), and Liver Weight

The GPT, GOT, and liver weights of OVX rats were markedly increased compared with those of sham rats (*p* = 0.002 for GPT; *p* = 0.03 for GOT; *p* = 0.02 for liver weight, Figure 4). After treatment with 100 mg/kg and 200 mg/kg PL, the GPT significantly reduced to levels similar to those in the sham (*p* = 0.002 and 0.004, respectively, Figure 4a). The administration of 50 mg/kg PL also reduced the GPT in the OVX + PL50, but the difference was not statistically significant with a Bonferroni correction (*p* = 0.03). The GOT also significantly decreased in the OVX + PL100 and OVX + PL200 (*p* = 0.0009 for both groups, Figure 4b). For the liver weight, we observed no significant changes in OVX + PL50, OVX + PL100, and OVX + PL200 with a Bonferroni correction (*p* = 0.02, 0.02, and 0.05, respectively, Figure 4c).

#### 3.1.4. Peroxisome Proliferator-Activated Receptor Gamma Messenger RNA Expression

The PPAR-γ mRNA expression in the thigh muscle decreased in the OVX group when compared with the sham (*p* = 0.002). The PPAR-γ mRNA expression of OVX + PL50 (*p* = 0.006), OVX + PL100 (*p* = 0.001), and OVX + PL200 (*p* = 0.002) was significantly increased, similar to that of the OVX + RC group (*p* = 0.001) (Figure 5).

### 3.2. Network Pharmacological Analyses

We performed network pharmacological analyses to determine PL’s potential bioactive compounds and their targets and to predict the underlying mechanisms of PL on menopause-related metabolic diseases.

#### 3.2.1. Selection of Potential Bioactive Compounds of *Pueraria lobata*

Eighteen PL compounds were retrieved from the TCMSP (Table 2). To evaluate the ADME (absorption, distribution, metabolism, and elimination or excretion), we considered the pharmacokinetic parameters of the compounds: oral bioavailability (OB) and drug likeness (DL). The OB represents the possibility of the drug reaching the target upon oral administration; the DL index evaluates whether a compound can be bioactive when compared to the known drugs. As a result, we identified four potential bioactive compounds satisfying the screening criteria (OB ≥ 30%, DL ≥ 0.18): formononetin, beta-sitosterol, 3′-methoxydaidzein, and daidzein-4,7-diglucoside.

#### 3.2.2. Target Analysis

With the potential bioactive compounds of PL, we identified 61 potential targets by using an in silico model from the TCMSP (Table 3). These targets are either experimentally validated or predicted by machine learning algorithms [28]. To identify the genes that are known to be associated with menopause among the 61 targets, we retrieved menopause-related genes from the Entrez Gene database. Four overlapping genes were found between the predicted 61 targets of PL’s potential bioactive compounds and 110 menopause-related genes from the Entrez Gene database: estrogen receptor 1 (ESR1), estrogen receptor 2 (ESR2), nitric oxide synthase 3 (NOS3), and beta-2 adrenoreceptor (ADRB2).

#### 3.2.3. Identifying Potential Pathways of *Pueraria lobata*

We performed KEGG pathway enrichment analysis to better understand the underlying pathways that are associated with the predicted 61 targets. A total of 107 enriched signaling pathways were obtained at an adjusted *p*-value of 0.05, among which 30 pathways with high combined scores are presented in Table 4. We found that four pathways were related to the female sex hormones in the top 30 pathways: estrogen signaling pathway, progesterone-mediated oocyte maturation, oxytocin signaling pathway, and prolactin signaling pathway. Menopause-related genes (except *ADRB2*) were also found to be involved in the female sex hormone-related pathways. These genes have been shown to be involved in an additional nine pathways that can be potential pathways by which PL relieves menopausal symptoms.

#### 3.2.4. Construction of the Compound—Target Network of *Pueraria lobata*

We constructed and visualized a bipartite compound—target network of PL to understand their interactions at the systems level. The network consisted of 65 nodes corresponding to the potential bioactive compounds or their targets and 109 edges indicating interactions between compounds and targets (Figure 6). Among the targets of formononetin, 3′-methoxydaidzein, beta-sitosterol, and daidzein-4,7-diglucoside, 15 targets were found to be related to female sex hormone-related pathways: *PIK3CG, PRKACA, JUN, NOS3, CALM1, MAPK14, ESR1, ESR2, HSP90AA1, PTGS2, CCNA2, CDK2, PGR, OPRM1*, and *GSK3B*. The numbers of related targets for the estrogen signaling pathway, progesterone-mediated oocyte maturation, oxytocin signaling pathway, and prolactin signaling pathway were 9, 7, 6, and 5, respectively.

## 4. Discussion

In this study, we investigated the multiple effects and underlying mechanisms of PL on menopause-related metabolic diseases. The administration of PL significantly improved the serum lipids, GOT, and GPT in OVX rats, implying that PL has potential as a therapeutic agent for menopause-related chronic diseases. Suppressed PPAR-γ expression in the OVX rats was significantly attenuated after treatment with PL. Additionally, network pharmacological analysis has led to the identification of bioactive compounds and their potential pathways, which suggests that the mechanism of PL might be exerted via multiple pathways related to female sex hormones. We expect that PL would have preventive effects on dyslipidemia and CVD after menopause.

In menopausal women, estrogen deficiency increases the risk of hyperlipidemia, CVD, and liver disease. This is related to the role of estrogen in lipid metabolism in which it inhibits serum low density lipoprotein (LDL) cholesterol, increases HDL cholesterol production, and inhibits homocysteine production and vascular function. Activation of estrogen receptors improves peripheral energy and glucose homeostasis in multiple ways. It prevents liver steatosis, suppresses hepatic glucose production, and improves insulin sensitivity [1]. It enhances lipid oxidation, insulin-sensitive glucose transporter GLUT4 expression, and insulin sensitivity in skeletal muscle [29].

Previous studies have shown elevated GOT, GPT, hepatocyte hypertrophy, and fat changes in the liver in OVX rats [30,31]. This study also supports the possibility that PL can be used for the treatment and prevention of menopause-related metabolic disorders. The levels of total cholesterol and triglyceride in the OVX model were significantly improved by PL administration, implying the possibility that PL could be used for the prevention of postmenopausal CVD and dyslipidemia. In addition, GPT and GOT were significantly decreased, confirming that PL showed a liver protective effect in the OVX model.

The PPAR-γ, analyzed as a common target of beta-sitosterol, formononetin, and 3′-methoxydaidzein, is a nuclear receptor whose genetic variants result in altered insulin sensitivity and lipid storage [32]. It is expressed at low levels in skeletal muscle, where it protects against adiposity and insulin resistance. The PPAR-γ action directly promotes myocellular storage of energy by increasing fatty acid uptake and esterification while simultaneously enhancing insulin signaling and glycogen formation [33]. Muscle-specific PPAR-γ knockout (Mu PPAR-γ KO) mouse studies have revealed the additional metabolic importance of skeletal muscle PPAR-γ as these mice exhibit elevated serum lipids and excess weight gain [34,35]. We observed that the expression of PPAR-γ was significantly increased in all PL administration groups regardless of dose. Therefore, our experimental results and network pharmacological analysis consistently suggest PPAR-γ as a candidate target of PL in lipid metabolism.

Using network pharmacological analyses, we identified four potential bioactive compounds of PL satisfying the criteria of OB and DL: formononetin, 3′-methoxydaidzein, beta-sitosterol, and daidzein-4,7-diglucoside. Formononetin has been reported to have estrogenic effects and has also been shown to improve antioxidant defense mechanisms through the enzymatic and nonenzymatic systems of tissues and to inhibit lipid peroxidation by eliminating free radicals [36]. Beta-sitosterol is known to be effective in hypercholesterolemia, heart disease, immune system regulation, and cancer prevention [37]. In particular, LDL cholesterol has been shown to act in a competitive manner to inhibit cholesterol absorption in the body, thereby reducing blood cholesterol and β-lipoprotein levels [38]. Additionally, it promoted estrogen-responsive breast cancer cell proliferation, acting similarly to estrogen [39]. 3′-Methoxydaidzein is reported to have estrogenic activity affecting breast cancer cells [40]. Daidzein-4,7-diglucoside is regarded to act as a phytoestrogen [41].

Among the 61 potential targets of PL, we retrieved four genes that are related to menopause: *ESR1*, *ESR2*, *NOS3*, and *ADRB2*. The accordance rate (four matched targets of 61 predicted targets) was higher than the chance level (*p*-value < 0.001, hypergeometric test), supporting the reliability of the predicted results. The *ESR1* and *ESR2*, which are involved in the estrogen signaling pathway, are highly related to menopause-related metabolic diseases [42]. A previous study has shown that *ESR1* and *NOS3* predict postmenopausal CVD-related endothelial responses and can be used to predict insulin resistance and type 2 diabetes risk [43]. The *ADRB2* is known to be an obesity-related gene and is associated with disturbances in β-adrenoceptor-mediated lipolysis and fat oxidation [44], so it may affect lipid metabolism after menopause.

## 5. Conclusions

We found that PL improved various indicators associated with lipid metabolism in the postmenopausal animal model and that it simultaneously acts on several targets related to menopause-related metabolic diseases. Our results suggested that the PL targets could change serum lipid levels and liver function indicators in the OVX model by affecting multiple pathways, including the estrogen-related pathway. We expect the network pharmacological method, which systematically analyzes the action of drugs, will play a role in presenting a new direction in research design and interpretation of the underlying mechanism of drugs in the future.

## Figures and Tables

**Figure 1 biomolecules-09-00747-f001:**
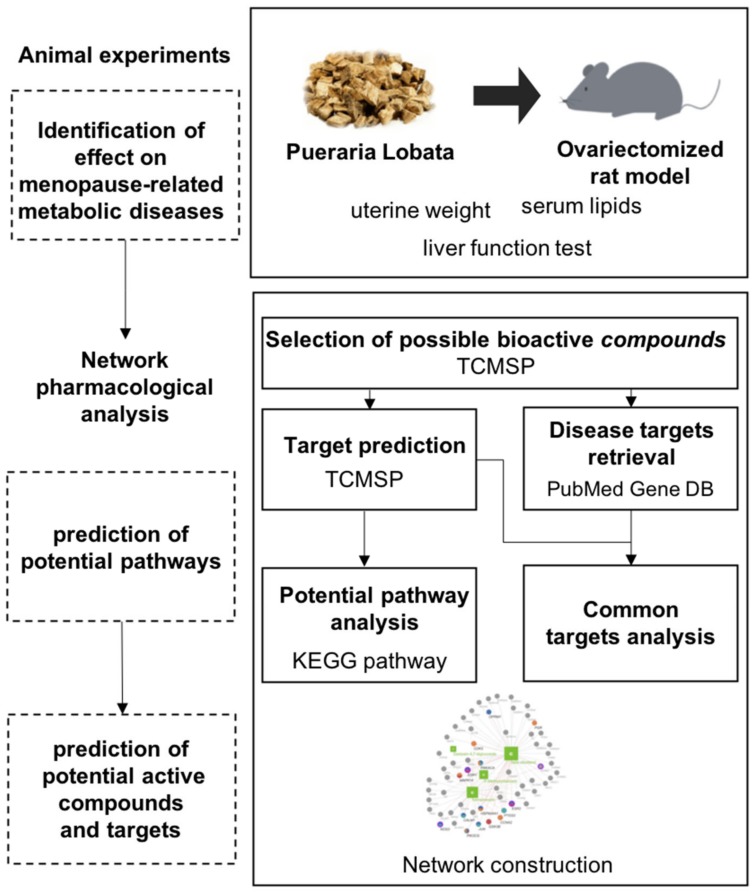
Overview of the study process. Abbreviations: TCMSP, Traditional Chinese Medicine Systems Pharmacology; KEGG, Kyoto Encyclopedia Genes and Genomes.

**Figure 2 biomolecules-09-00747-f002:**
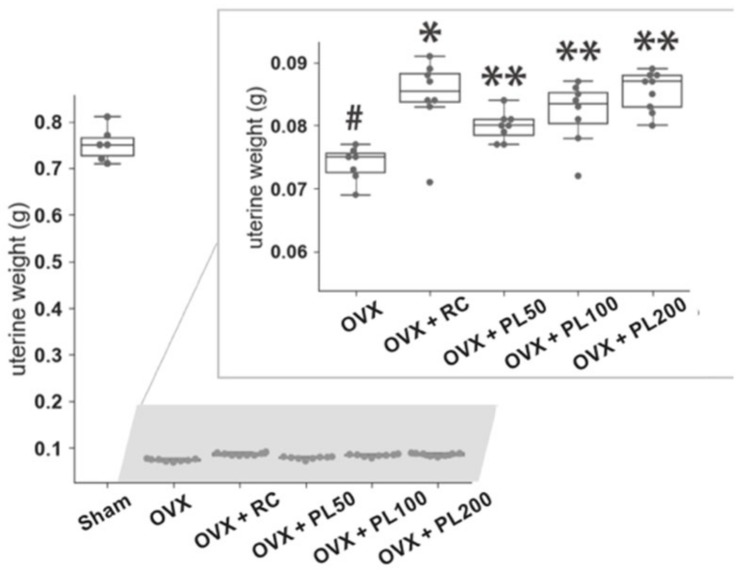
The effect of PL on the uterine weight of OVX rats. Zoomed graph is the data except for the sham group. Each point indicates the value of each animal. In the box plots, the upper and lower boundaries of the box mark the 75th percentile and the 25th percentile, respectively. A line within the box indicates the median, and whiskers above and below the box indicate 1.5 interquartile range (75th percentile–25th percentile). Points outside the box are identified as outliers. The results were compared by the Mann–Whitney U test; sham vs. OVX: #*p* ≤ 0.05; OVX vs. OVX + RC: **p* ≤ 0.05; OVX vs. OVX + PL: **adjusted *p* ≤ 0.05.

**Figure 3 biomolecules-09-00747-f003:**
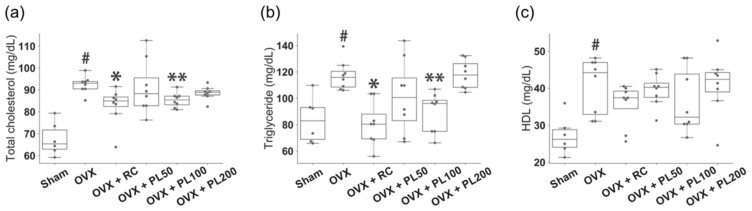
The effect of PL on serum lipids in OVX rats. (**a**) Total cholesterol; (**b**) triglyceride; (**c**) HDL; For description of the box plots, please refer to Figure 2. The results compared by the Mann-Whitney U test; sham vs. OVX: #*p* ≤ 0.05; OVX vs. OVX + RC: **p* ≤ 0.05; OVX vs. OVX + PL: **adjusted *p* ≤ 0.05.

**Figure 4 biomolecules-09-00747-f004:**
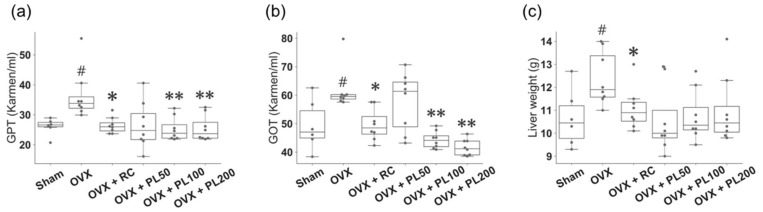
The effect of PL on liver function and liver weight of OVX rats. (**a**) GPT; (**b**) GOT; (**c**) liver weight. For description of the box plots, please refer to Figure 2. The results compared by the Mann–Whitney U test; sham vs. OVX: #*p* ≤ 0.05; OVX vs. OVX + RC: **p* ≤ 0.05; OVX vs. OVX + PL: **adjusted *p* ≤ 0.05.

**Figure 5 biomolecules-09-00747-f005:**
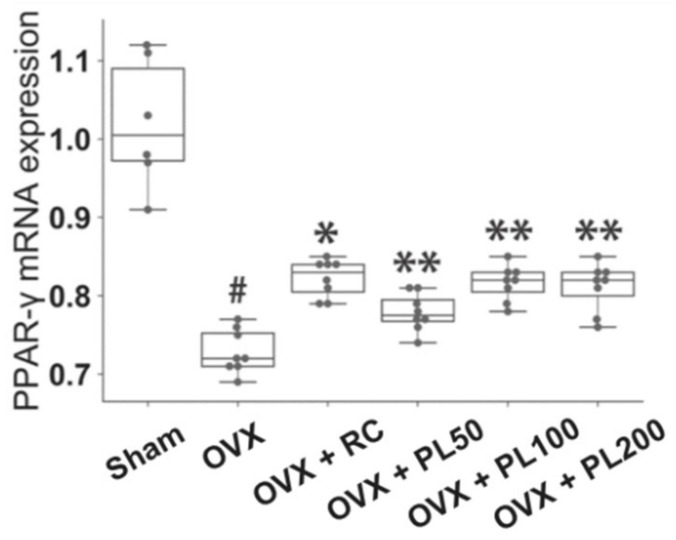
The effect of PL on PPAR-γ mRNA expression in OVX rats. For description of the box plots, please refer to Figure 2. The results compared by the Mann–Whitney U test; sham vs. OVX: #*p* ≤ 0.05; OVX vs. OVX + RC; **p* ≤ 0.05; OVX vs. OVX + PL: **adjusted *p* ≤ 0.05.

**Figure 6 biomolecules-09-00747-f006:**
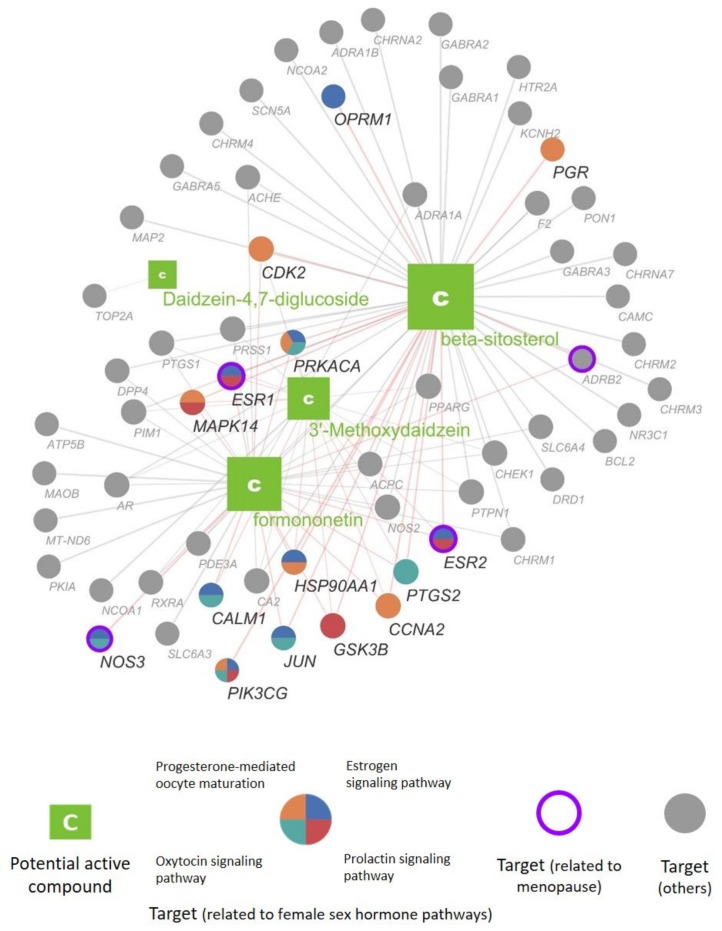
Compound–target network of PL. Each node represents a compound or target (explained in the box), and the edge indicates the interaction between the compound and the target. The size of each node is proportional to its degree (the number of edges). Genes associated with menopause or female sex hormone-related pathways are highlighted with different colors in the network.

**Table 1 biomolecules-09-00747-t001:** Primer sequence used for polymerase chain reaction (PCR).

Primer	Sense (5′-3′)	Antisense (5′-3′)
Peroxisome proliferator-activated receptor gamma (PPAR-γ)	TCPLAPLGCTCTGTCATC	CATCTGTACTPLTPLPLACA

**Table 2 biomolecules-09-00747-t002:** Compounds of *Pueraria lobata* (PL) with their oral bioavailability (OB), drug likeness (DL), and molecular properties.

Compound Name	MW	OB (%)	DL	AlogP	Hdon	Hacc	Caco-2
Formononetin ^*^	268.28	69.67	0.21	2.58	1	4	0.78
Sitogluside	576.95	20.63	0.62	6.34	4	6	−0.14
Beta-sitosterol ^*^	414.79	36.91	0.75	8.08	1	1	1.32
Daidzein	254.25	19.44	0.19	2.33	2	4	0.59
Ononin	430.44	11.52	0.78	0.68	4	9	−0.74
Docosanoate	340.66	15.69	0.26	9.11	1	2	1.21
Lupenone	424.78	11.66	0.78	7.36	0	1	1.48
Genistein	270.25	17.93	0.21	2.07	3	5	0.43
Lignoceric acid	368.72	14.9	0.33	10.02	1	2	1.24
Scoparone	206.21	74.75	0.09	1.87	0	4	0.85
(R)-allantoin	158.14	96.9	0.03	−1.76	5	7	−0.99
3′-Methoxydaidzein ^*^	284.28	48.57	0.24	2.32	2	5	0.56
Daidzein-4,7-diglucoside ^*^	578.57	47.27	0.67	−1.48	8	14	−2.53
Soyasapogenol b	458.8	16.73	0.75	5.11	3	3	0.43
Puerarin	416.41	24.03	0.69	−0.06	6	9	−1.15
7,8,4′-Trihydroxyisoflavone	270.25	20.67	0.22	2.07	3	5	0.45
Daidzin	416.41	14.32	0.73	0.43	5	9	−1
Sophoradiol	442.8	17.42	0.76	6.2	2	2	0.95

Note that compounds marked with an * satisfy the screening criteria (OB ≥ 30, DL ≥ 0.18). MW, molecular weight; AlogP, octanol-water partition coefficient log P; Hdon, hydrogen bond donor; Hacc, hydrogen bond acceptor; Caco-2, Caco-2 permeability.

**Table 3 biomolecules-09-00747-t003:** Predicted targets of the potential bioactive compounds of *Pueraria lobata* (61 genes).

Predicted Targets of the Potential Bioactive Compounds of PL								
*ESR1*	*ESR2*	*JUN*	*MT-ND6*	*ATP5B*	*NOS2*	*PTGS1*	*F2*	*CHRM1*
*AR*	*PPARG*	*PTGS2*	*NOS3*	*CA2*	*RXRA*	*ACHE*	*PDE3A*	*ADRA1A*
*PTPN1*	*SLC6A3*	*ADRB2*	*SLC6A4*	*DPP4*	*MAPK14*	*GSK3B*	*HSP90AA1*	*CDK2*
*ACPC*	*MAOB*	*CHEK1*	*PRKACA*	*PRSS1*	*PIM1*	*CCNA2*	*PKIA*	*CALM1*
*BCL2*	*PON1*	*MAP2*	*DRD1*	*CHRM3*	*KCNH2*	*SCN5A*	*GABRA2*	*CHRM4*
*HTR2A*	*GABRA5*	*GABRA3*	*PGR*	*CHRM2*	*ADRA1B*	*CHRNA2*	*OPRM1*	*NR3C1*
*GABRA1*	*PIK3CG*	*CHRNA7*	*CAMC*	*NCOA2*	*NCOA2*	*TOP2A*		

**Table 4 biomolecules-09-00747-t004:** KEGG pathway enrichment analysis of potential target genes of PL’s potential bioactive compounds (adjusted *p*-value ≤ 0.05).

Pathway	Combined Score	Adjusted *p*-Value	Genes (Targets)
Neuroactive ligand-receptor interaction ^†^	88.66	4.45 × 10^−11^	*GABRA2;CHRM2;GABRA1;CHRM3;PRSS1;CHRNA2;CHRM1;GABRA5;CHRM4;CHRNA7;GABRA3;ADRB2;OPRM1;HTR2A;F2;ADRA1B;NR3C1;ADRA1A;DRD1*
Calcium signaling pathway ^†^	62.88	1.39 × 10^−7^	*CHRM2;CHRM3;CHRM1;NOS2;NOS3;CHRNA7;ADRB2;HTR2A;ADRA1B;ADRA1A;DRD1;CALM1;PRKACA*
Cholinergic synapse	46.9	1.19 × 10^−5^	*CHRM2;CHRM3;ACHE;CHRM1;CHRM4;CHRNA7;BCL2;PRKACA;PIK3CG*
Estrogen signaling pathway *^†^	46.29	6.38 × 10^−6^	*HSP90AA1;JUN;NOS3;OPRM1;CALM1;PRKACA;ESR1;ESR2;PIK3CG*
Pathways in cancer	38.78	6.59 × 10^−4^	*AR;GSK3B;JUN;HSP90AA1;RXRA;NOS2;CDK2;BCL2;PPARG;PRKACA;PTGS2;PIK3CG*
Morphine addiction	38.13	2.63 × 10^−5^	*GABRA2;GABRA1;GABRA5;GABRA3;PDE3A;OPRM1;DRD1;PRKACA*
Adrenergic signaling in cardiomyocytes ^†^	36.69	7.71 × 10^−5^	*BCL2;ADRB2;SCN5A;MAPK14;CALM1;PRKACA;ADRA1B;ADRA1A;PIK3CG*
cAMP signaling pathway ^†^	32.7	4.14 × 10^−4^	*CHRM2;JUN;CHRM1;PDE3A;ADRB2;DRD1;CALM1;PRKACA;PIK3CG*
Progesterone-mediated oocyte maturation *	30.5	3.51 × 10^−4^	*CCNA2;HSP90AA1;CDK2;PGR;MAPK14;PRKACA;PIK3CG*
Retrograde endocannabinoid signaling	29.53	3.54 × 10^−4^	*GABRA2;GABRA1;GABRA5;GABRA3;MAPK14;PRKACA;PTGS2*
Amphetamine addiction	27.59	3.54 × 10^−4^	*JUN;MAOB;DRD1;CALM1;PRKACA;SLC6A3*
Thyroid hormone signaling pathway ^†^	25.81	6.36 × 10^−4^	*NCOA1;NCOA2;GSK3B;RXRA;PRKACA;ESR1;PIK3CG*
Dopaminergic synapse	24.97	7.51 × 10^−4^	*GSK3B;MAOB;DRD1;MAPK14;CALM1;PRKACA;SLC6A3*
Epstein–Barr virus infection	24.92	1.47 × 10^−3^	*CCNA2;GSK3B;JUN;CDK2;BCL2;MAPK14;PRKACA;PIK3CG*
Taste transduction	22.57	6.59 × 10^−4^	*GABRA2;GABRA1;CHRM3;GABRA5;GABRA3;PRKACA*
Prostate cancer	22.07	7.51 × 10^−4^	*GSK3B;AR;HSP90AA1;CDK2;BCL2;PIK3CG*
AGE-RAGE signaling pathway in diabetic complications ^†^	21.27	1.32 × 10^−3^	*JUN;NOS3;PIM1;BCL2;MAPK14;PIK3CG*
Small cell lung cancer	21.1	7.45 × 10^−4^	*RXRA;NOS2;CDK2;BCL2;PTGS2;PIK3CG*
PI3K-Akt signaling pathway ^†^	20.52	7.80 × 10^−3^	*CHRM2;GSK3B;HSP90AA1;CHRM1;RXRA;NOS3;CDK2;BCL2;PIK3CG*
Salivary secretion ^†^	19.81	7.51 × 10^−4^	*CHRM3;ADRB2;CALM1;ADRA1B;PRKACA;ADRA1A*
cGMP-PKG signaling pathway ^†^	19.71	2.42 × 10^−3^	*NOS3;PDE3A;ADRB2;CALM1;ADRA1B;ADRA1A;PIK3CG*
Nicotine addiction	18.71	3.80 × 10^−4^	*GABRA2;GABRA1;GABRA5;CHRNA7;GABRA3*
Serotonergic synapse	16.7	2.05 × 10^−3^	*MAOB;HTR2A;PRKACA;PTGS2;SLC6A4;PTGS1*
Neurotrophin signaling pathway	16.19	2.59 × 10^−3^	*GSK3B;JUN;BCL2;MAPK14;CALM1;PIK3CG*
Regulation of lipolysis in adipocytes ^†^	15.31	8.42 × 10^−4^	*ADRB2;PTGS2;PRKACA;PIK3CG;PTGS1*
Cocaine addiction	14.94	6.59 × 10^−4^	*JUN;MAOB;DRD1;PRKACA;SLC6A3*
Oxytocin signaling pathway *^†^	14.46	8.60 × 10^−3^	*JUN;NOS3;CALM1;PRKACA;PTGS2;PIK3CG*
Prolactin signaling pathway *^†^	14.06	2.12 × 10^−3^	*GSK3B;MAPK14;ESR1;ESR2;PIK3CG*
Inflammatory mediator regulation of TRP channels	13.19	6.91 × 10^−3^	*HTR2A;MAPK14;CALM1;PRKACA;PIK3CG*
Alcoholism	12.61	1.34 × 10^−2^	*MAOB;PKIA;DRD1;CALM1;PRKACA;SLC6A3*

† indicates that the pathway is related to the overlapping genes between the predicted targets of PL and menopause-related genes retrieved from the Entrez Gene database; * indicates that the pathway is related to sex hormones.

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
