# Peer review of "Investigating the Systems-Level Effect of Pueraria lobata for Menopause-Related Metabolic Diseases Using an Ovariectomized Rat Model and Network Pharmacological Analysis"

_biomolecules, 2019, doi:10.3390/biom9110747_

Round 1

Reviewer 1 Report

The research has been correctly designed, it concerns an important scientific issue, the obtained results may also have practical application. The manuscript requires minor corrections regarding description of the methods and results.

In the ‘Introduction’ section, please provide information on the prevalence of metabolic syndrome in postmenopausal women.

Figure 1, explain the abbreviations used.

In the description of the methods, wherever reagents, laboratory equipment, computer programmes, etc. are mentioned, please provide the name of the manufacturer, city and country.

Please, combine information from Lines 80-82 and 110-113, the description of T-CHOL, HDL, TG, AST, ALT markings should be specified by providing test catalogue numbers and measurement characteristics, e.g. detection limit and measurement linearity, and
for the ELISA test, please provide the catalogue number, detection range, intra- and inter-assay CV, please, explain how the correctness of the measurement was verified (e.g. control samples of known concentration).

In the description of the results, please indicate which comparison is subject to statistical evaluation and whether the change was statistically significant providing a specific P value (not only P≤0.05). In the description of the figures, please, provide a way of presenting the results, which means box, mustache, horizontal line.

Line 159 - wrong header

The description of the results should not include fragments of discussion, such as lines 160, 171, 174. The information from lines 186-187 is a repetition of the description of the methods. This should be removed from the description of the results.

Line 197 - expand the ADME abbreviation.

Line 210 - use only TCMSP abbreviation, the expansion of the abbreviation was given earlier.

Delete lines 322-324.

Check and correct the list of references in accordance with the guidelines.

Author Response

Response to Reviewer 1 Comments

Point 1: The research has been correctly designed, it concerns an important scientific issue, the obtained results may also have practical application. The manuscript requires minor corrections regarding description of the methods and results.

In the ‘Introduction’ section, please provide information on the prevalence of metabolic syndrome in postmenopausal women.

Response 1: Thank you for the opinion. We added “the prevalence of metabolic syndrome in postmenopausal women.”
(Modified Line [48]: The prevalence of the metabolic syndrome in post-menopausal women is higher than in pre-menopausal women, which is estimated at 31-55% according to the various studies [3–6].)

Point 2: Figure 1, explain the abbreviations used. 

Response 2: Thank you for the correction. We added the explanation of abbreviations. Additionally, we also corrected TCMSP website url in the line 136 (revised manuscript) because they recently changed the url.
(Modified Line [84]: Abbreviations: OVX, ovariectomized; TCMSP, Traditional Chinese Medicine Systems Pharmacology; KEGG, Kyoto Encyclopedia Genes and Genomes.
(Modified Line [146]: The PL compounds were retrieved from the Traditional Chinese Medicine Systems Pharmacology (TCMSP, http://tcmspw.com/tcmsp.php)

Point 3: In the description of the methods, wherever reagents, laboratory equipment, computer programmes, etc. are mentioned, please provide the name of the manufacturer, city and country

Response 3: As you mentioned, we provided information of reagents, laboratory equipment. Every computer program we utilized was the open-source software platform, so we provided the website url.

Point 4: Please, combine information from Lines 80-82 and 110-113, the description of T-CHOL, HDL, TG, AST, ALT markings should be specified by providing test catalogue numbers and measurement characteristics, e.g. detection limit and measurement linearity, and
for the ELISA test, please provide the catalogue number, detection range, intra- and inter-assay CV, please, explain how the correctness of the measurement was verified (e.g. control samples of known concentration).

Response 4: The separated information for serum analysis have combined to section 2.4. The catalog number and manufacture’s information have included. The kits we used were measured within the range of the dosing line using the appropriate standard according to the manufacturer's protocol.

Point 5: In the description of the results, please indicate which comparison is subject to statistical evaluation and whether the change was statistically significant providing a specific P value (not only P≤0.05). In the description of the figures, please, provide a way of presenting the results, which means box, mustache, horizontal line.

Response 5: Thank you for your opinion. We indicated p-values and the statistical analyses we used. We also added the lines that explain the box plots.
(Modified Line [170]: Each point indicates the value of each animal. In the box plots, the upper and lower boundaries of the box mark the 75th percentile and the 25th percentile, respectively. A line within the box indicates the median, and whiskers above and below the box indicate 1.5 interquartile range (75th percentile-25th percentile). Points outside the box is identified as outliers. The results compared by the Mann-Whitney U test; sham vs OVX: #P≤0.05; OVX vs OVX+RC: *P≤0.05; OVX vs OVX+PL: **adjusted P≤0.05.)

Point 6: Line 159 - wrong header

Response 6: Thank you for the correction. We changed the header.
(Modified Line [176]: 3.1.2 Serum lipids)

Point 7: The description of the results should not include fragments of discussion, such as lines 160, 171, 174. The information from lines 186-187 is a repetition of the description of the methods. This should be removed from the description of the results.

Response 7: Thank you for the comment. We deleted the description of the results you’ve mentioned.

Point 8: Line 197 - expand the ADME abbreviation.

Response 8: Thank you for the correction. We expanded the ADME abbreviation.
(Modified [218]: ADME (absorption, distribution, metabolism and elimination or excretion)

Point 9: Line 210 - use only TCMSP abbreviation, the expansion of the abbreviation was given earlier.

Response 9: Thank you for the correction. We changed the word as TCMSP.

Point 10: Delete lines 322-324.

Response 10: Thank you. We deleted the lines.

Point 11: Check and correct the list of references in accordance with the guidelines

Response 11: Yes, we’ve checked the list of references.

Reviewer 2 Report

Authors describe the effects of Pueraria lobata on postmenopausal metabolic and hepatic profiles in the rat model. The study is well designed even if some changes are required throughout the paper

line 44 menopause syndrome menopausal symptoms line 50 HRT is not a preventive treatment of menopausal syndrome, but it is a treatment of menopausal symptoms line 53-54 is very inaccurate: long term effects of HRT such breast cancer are largely debated in the literature, but starting this sentence with the risk of irregular bleeding or endometrial hyperplasia seems inaccurate line 54-57 even if herbs can be beneficial research literature comparing natural products to placebo should be cited line 59 postmenopausal syndrome menopausal symptoms line 70 I do not believe that you aimed to assess the impact of PL on postmenopausal symptoms (hot flashes, sleep and mood disturbances, genitourinary syndrome) because you assessed metabolic parameters, hepatic function and so on but not menopausal symptoms. You should be more clear in your aims definitions line 75 you are investigating genes related to metabolism ecc not to menopausal symptoms. I suggest not to use the term menopausal syndrome as it is no appropriate, not coded. Menopause is not a syndrome! But menopause can bring bothersome symptoms. I recommend checking terminology throughout the paper line 102 they are not menopausal symptoms, but metabolic variations potentially associated with menopause line 159 error in the title (copy-paste from line 151) line 169 which is are line 249 same comments of line 75 (also apply to the abstract)

Author Response

Response to Reviewer 2 Comments

Point 1: Authors describe the effects of Pueraria lobata on postmenopausal metabolic and hepatic profiles in the rat model. The study is well designed even if some changes are required throughout the paper
line 44 menopause syndrome menopausal symptoms 

Response 1: Thank you for your comment. We have made the following modifications.
(Modified Line [44]: menopause syndrome à menopause related symptoms)

Point 2: line 50 HRT is not a preventive treatment of menopausal syndrome, but it is a treatment of menopausal symptoms. line 53-54 is very inaccurate: long term effects of HRT such breast cancer are largely debated in the literature, but starting this sentence with the risk of irregular bleeding or endometrial hyperplasia seems inaccurate. line 54-57 even if herbs can be beneficial research literature comparing natural products to placebo should be cited 

Response 2: Thank you for the comment. We agreed your opinion, so we modified the paragraph about HRT and non-hormonal therapy on menopausal disabling symptoms.
(Modified Line [52-63]: Hormone replacement therapy (HRT) is used as the first choice for treating different menopausal disabling symptoms. While HRT is effective in alleviating most post-menopausal symptoms [7],  many women refuse HRT concerning the latent adverse effects and prefer the use of non-hormonal therapies to relieve symptoms after menopause [8]. As one of the non-hormonal treatments, natural products are considered as an alternative to menopausal symptoms and metabolic diseases caused by menopause. In particular, herbs have been widely used to relieve symptoms after menopause in East Asia, especially in Korea, China, and Japan [9,10] Herb-based therapies were found to be associated with reductions in the symptoms in menopausal women than the placebo group in meta-analysis [10]. In a randomized controlled trial, whole soy had a beneficial effect on cardiovascular biomarkers in equol-producing post-menopausal women when compared with the placebo group. [11]. Several experimental studies reported that traditional herbal prescriptions reduced the serum lipid levels and hepatic lipid accumulation in an ovariectomized rat model [12,13].)

Point 5: line 59 postmenopausal syndrome menopausal symptoms. line 70 I do not believe that you aimed to assess the impact of PL on postmenopausal symptoms (hot flashes, sleep and mood disturbances, genitourinary syndrome) because you assessed metabolic parameters, hepatic function and so on but not menopausal symptoms. You should be more clear in your aims definitions. line 75 you are investigating genes related to metabolism ecc not to menopausal symptoms. I suggest not to use the term menopausal syndrome as it is no appropriate, not coded. Menopause is not a syndrome! But menopause can bring bothersome symptoms. I recommend checking terminology throughout the paper. line 102 they are not menopausal symptoms, but metabolic variations potentially associated with menopause. line 169 which is are line 249 same comments of line 75 (also apply to the abstract)

Response 5: Thank you for the good point. We agreed that the indexes we observed are related to metabolic diseases after menopause. As you mentioned, we changed all terminologies ‘menopause syndrome’, ‘menopausal symptoms’, and ‘postmenopausal symptoms’ into ‘menopause-related metabolic diseases’ throughout the paper. We also corrected our research objectives clearer: ‘we aimed to investigate the potential benefits of PL on menopause-related metabolic diseases at the systems-level.’

Point 6: line 159 error in the title (copy-paste from line 151) 

Response 6: Thank you for the correction. We changed the header.
(Modified Line [176]: 3.1.2 Serum lipids)

Round 2

Reviewer 2 Report

The authors modified the paper adequately and consistently with my suggestions.